# Restorative Effects of Classroom Soundscapes on Children’s Cognitive Performance

**DOI:** 10.3390/ijerph16020293

**Published:** 2019-01-21

**Authors:** Shan Shu, Hui Ma

**Affiliations:** School of Architecture, Tianjin University, Tianjin 300072, China; shan.shu@outlook.com

**Keywords:** restorative effect, children’s cognitive performance, classroom soundscape, sustained attention, short-term memory

## Abstract

Previous studies have examined the restorative benefits of soundscapes on adults’ cognitive performance, but it was unclear whether those benefits would be possible for children. In this paper, two experiments applied a before–after design to explore the restorative effects of different soundscapes on children’s sustained attention and short-term memory, respectively, in a simulated classroom situation. In Experiment 1, 46 children aged 8–12 were first mentally fatigued by performing an oral arithmetic task and then were asked to conduct a sustained attention to response test (SART), in order to assess their attention fatigue. After that, a period of 3-min soundscape was presented, and SART was conducted again to examine their attention recovery. In Experiment 2, 45 children participated and the experiment procedure was the same as in Experiment 1, except that a digit span test (DST) was used instead to measure short-term memory. The results showed that music, birdsong, fountain sound, and stream sound facilitated greater recovery than other sounds in reaction time. Participants also showed better performance in short-term memory after exposure to fountain sound and stream sound, followed by music and birdsong. Those results confirmed the actual restorative effects of perceived restorative soundscapes on children’s cognitive performance.

## 1. Introduction

Numerous studies have documented the positive effects of interacting with natural environments [1]. As an essential part of physical environments, emerging research has demonstrated that some pleasant soundscapes might have potential benefits on people’s well-being and health [2,3,4].

The restorative benefits of soundscapes could be framed in terms of two prevailing theories of restorative environments: (1) Stress recovery theory (SRT) [5] and (2) attentional restoration theory (ART) [6]. SRT focuses primarily on the effects of environmental stimuli on emotional and physical responses, suggesting that interacting with nature could evoke positive responses indicative of stress reduction [7]. Indeed, numerous studies have provided some evidence for this by demonstrating an increase in pleasant mood [8], a decrease in heart rate [9,10,11], a reduced skin conductance level [12], and other stress-relieving measures. ART, which is commonly referenced to identify and restore a cognitive mechanism [13], draws on research demonstrating that attention can be separated into two components: Involuntary attention, in which attention is captured by salient stimuli, and voluntary or directed attention, in which attention is directed by cognitive-control processes [14]. According to ART, exposure to restorative stimuli with inherently fascinating quality could invoke involuntary attention, thereby allowing directed attention mechanisms a chance to replenish. Therefore, following an interaction with restorative stimuli, one is able to perform better on cognitive tasks that depend on directed attention abilities. This theory has been supported by numerous studies, including in-situ studies [15,16,17] and laboratory experiments [18], which focused on the comparison of various natural and urban scenes. However, there is still some uncertainty regarding which aspects of attention may be affected by exposure to restorative stimuli [19].

In accordance with ART, cognitive benefits from soundscape exposure have been frequently reported based on human perceptions. Most studies investigated the restorative qualities of soundscapes in terms of cognitive restoration using questionnaires. For instance, based on the four components that are highlighted in ART (fascination, being away, compatibility, and extent), which were important for producing a restorative environment, Payne developed a perceived restorativeness soundscape scale (PRSS) to assess perceptions of a soundscape’s restorative potential [20] and further explored its construct validity [21]. This scale successfully differentiated the restorativeness of soundscapes from different urban parks and was thus used by other researchers [3,22]. In addition, a study involving semistructured interviews found that bird sounds were perceived as welcome distractions that effortlessly removed participants from cognitive fatigue [23]. Another laboratory experimental study showed a positive change in patients’ cognitive response (i.e., interest and understanding) after exposure to hospital ward soundscape clips combined with bird singing and babbling stream [24]. A review paper referring to sound interventions in the clinic environment also suggested that pleasant natural sound intervention, which includes singing birds, gentle wind, and ocean waves, had benefits that contributed to perceived restoration of attention in patients and staff [25]. Therefore, the restorative potentials of soundscapes on cognition have been widely reported by previous research.

However, those studies were mainly based on the self-reported experience of cognitive restoration. Empirical studies involving cognitive benefits of sounds found mixed results: A few studies reported actual recovery of cognitive functioning after soundscape exposure [26,27], while others demonstrated no significant effects of restorative soundscapes on cognitive measures [28,29,30]. For example, a study used digit span backwards as a measure of cognitive performance to compare the restorative effects of different sounds and found that participants exposed to natural sounds did not perform better than those who were exposed to anthropogenic sounds [31]. Similarly, another laboratory study measured participants’ task performance before and after sound exposure. Although results showed all sounds had positive effects on task performance, there was no significant difference between noise and natural sounds [32]. Those previous empirical studies applied quite different experiment stimuli, presentation methods, and measurements, thereby resulting in inconsistent outcomes. However, it is noteworthy that the research using audiovisual stimuli usually resulted in more restorative effects than those using only sound stimuli. In addition, various cognitive tests regarding attention and memory were mainly used as sensitive measures of cognitive restoration. Overall, even if some research has indicated the potential restorative effects of soundscapes, this theory still needs to be adequately tested by more evidence-based research with a systematic approach.

To date, however, not only does the evidence of soundscapes’ cognitive benefits remain inconsistent, but it is also not clear whether these benefits could be generalized to children, since previous research primarily focused on adults. Given the widely reported academic burden in primary schools of China, which imposes increasing demands on children’s cognitive resources and leads to fatigue and decline in cognitive performance, it may be a quite urgent task to explore the restorative stimuli that could facilitate cognitive recovery for children [33]. The cognitive capabilities needing restoration therein encompass sustained attention and short-term memory, which were often used as cognitive indicators of directed attention highlighted in ART [15,18]. Those two basic cognitive abilities are believed to underlie the emergence of a more complex cognitive progress and play an important part in children’s education processes, such as learning, reading, and writing [34,35]. Therefore, these two cognitive capabilities should be considered first for investigation in the study of children’s cognitive restoration.

Moreover, in contrast with annoyed ambient noise, which has been widely proven to have adverse effects on the two cognitive abilities in prior studies [36,37], pleasant soundscapes exhibited restorative potentials as perceived by children [22]. Thus, it seems reasonable to expect a positive effect on children’s cognition after exposure to a restorative soundscape. In addition, considering the rapid development of children’s cognitive functioning, which is less automatized and more easily influenced by surrounding environments compared to that of adults [38], the effects of restorative soundscapes on cognitive performance are likely to be different between children and adults.

The aim of this study was to explore the restorative benefits of various soundscapes on children’s cognitive performance. Given that classrooms serve as a major educational environment for children’s cognitive development [39], this study was carried out in a simulated classroom situation. Specifically, two experiments were carried out to examine the cognitive benefits of soundscapes: In Experiment 1, a sustained attention to response test (SART) was used as a measure of sustained attention, and in Experiment 2, a digit span test (DST) was used as a measure of short-term memory. We hypothesized that exposure to different classroom soundscapes has different restorative effects on children’s sustained attention and short-term memory.

## 2. Materials and Methods

### 2.1. Participants

A total of 95 children aged 8–12 were recruited as participants in this study. Specifically, 46 children (mean = 10.25 years, SD = 1.33) participated in Experiment 1 (SART), and 45 children (mean = 10.31 years, SD = 1.40) participated in Experiment 2 (DST). Participants were recruited via social media and snowball sampling, from various primary schools in Tianjin, China. All the participants reported that they had normal hearing and normal vision. In addition, four participants were excluded during the data analysis, because their baseline cognitive scores were outliers.

The study was conducted in accordance with the Declaration of Helsinki. Ethical approval was obtained from the Academic Committee of School of Architecture, Tianjin University. All the children and their parents were informed about the study protocol, and they voluntarily participated in the study. Before the experiment, children gave oral consent, and parents gave written informed consent to the researchers.

In order to investigate the influence of demographic characteristics on children’s cognitive restoration, participants were regrouped to simplify data analysis and interpretation. The variable ‘age’ was categorized to indicate the ranges: 7–8 years, 9–10 years, and 11–12 years. The baseline of CE (commission errors) was categorized to two levels: Fewer (≤9) and more (≥10) errors. The baseline of RT (reaction time) was categorized to three levels: Fast (<450), medium (450–500), and slow (>500). The baseline of DS (digit span) was categorized to two levels: Short (≤9) and long (≥10) sequence. Participants were grouped in order to have a similar sample size of each group. Sample demographics are shown in Table 1.

### 2.2. Experimental Stimuli

In order to provide a complete and realistic presentation of a classroom and avoid other distractions in the lab, an immersive virtual reality (VR) technology was employed in this study. A visual recording of the classroom was captured in a typical primary school of Tianjin, China. A panoramic cameral consisting of six cameras (Insta360 Pro) arranged around a sphere was used to capture an omnidirectional picture of the classroom. The camera was placed on a tripod in the middle of the classroom with a height of 1.0m from the ground, to simulate the view of children when they were sitting. The panorama of the classroom was played in a VR head-mounted display (HMD) during the experiments, as shown in Figure 1.

In this study, five environmental sounds (i.e., music, birdsong, fountain sound, bell ring, and stream sound) and ambient noise in the environment of classrooms were used to generate the sound stimuli. The five environmental sounds were selected for the following reasons: (1) They were assessed by children as the potential restorative sounds in a prior study [22] and (2) they are coherent with the classroom setting and could be easily added in a school classroom through corresponding soundscape design. Specifically, a typical classical piano music, named “Souvenirs d’ enfance”, was selected as the music stimulus, which was a piece of soft and relaxing music without lyrics. In addition, a series of chirps of sparrows and other few bird species were recorded in a park in Tianjin and used as the birdsong stimulus. The fountain sound was produced by an upward shallow jet and water falling along all sides of a square stone column, while the stream sound was recorded nearby a downwards stream consisting of slightly tilted stone steps and slowly flowing water. The sound of the bell ring was produced by irregularly striking a wind chime, which was constructed from a series of suspended mental bells. As for the ambient noise of the classroom, it was recorded in an unoccupied classroom with the window closed. The ambient noise included environmental sounds outside the classroom and noise from construction equipment in the building. The sound recordings were all collected over dual channels using a Sony PCM-D50 digital recorder equipped with two stereo microphones (set at 16 bit and 44,100 Hz sampling rate). A 30-s sample was extracted from each of the six sound recordings for audio reproduction.

To simulate the real-life settings of classrooms, five environmental sounds (i.e., music, birdsong, fountain sound, bell ring, and stream sound) were used as dominant sound signals, which were separately combined with the ambient noise of classrooms to generated five soundscape stimuli. Therefore, a preliminary test was carried out to identify the optimal restorative signal-to-noise (S/N) of dominant sounds and ambient noise. In the test, ambient noise in the classroom was set as 45 dBA according to the upper limit of noise standard in China primary school classrooms, and the S/N of five levels (−5, 0, 5, 10, 15) was used to simulate the sound environments in classrooms. Thirty children aged 8–12 were exposed to the soundscapes, and they were asked to evaluate each soundscape using a perceived restorative soundscape scale for children [22]. Results showed that children reported an S/N of 5 dB to be most restorative. Therefore, in this study, the A-weighted equivalent sound pressure level of five environmental sounds was set as 50 dBA, while the ambient noise was set as 45 dBA, with an S/N of 5 dB. Before the combination, the sound signals were normalized to the average sound level of 50 dBA (45 dBA for ambient noise) using a Norsonic Nor140 Class 1 sound level meter coupled to a reference class headphone AKG K702, whose high sensitivity could ensure the accuracy of the emitted signal with respect to the real sound. Both the adjustment of sound pressure level and the combination of soundscapes were processed by Adobe Audition software.

The psychoacoustic characteristics and restorative evaluation of the sound signals and combined soundscapes were analyzed using the Artemis software (HEAD acoustics), as shown in Table 2. As reported in a previous study, the Just Noticeable Differences (JNDs) of loudness, fluctuation strength, sharpness, and roughness were 0.5 sone, 0.012 vacil, 0.08 acum, and 0.04 asper, respectively [40]. It could be seen that the psychoacoustic values were quite scattered according to the types of sound signals and soundscapes. Thus, it was expected that the participants can perceive the difference of those parameters among soundscapes.

Altogether, six soundscapes, including music, birdsong, fountain sound, bell ring, and stream sound and ambient noise, were used as experimental sounds, while silence was used as a control stimulus in this study. Silence was widely reported to be an important factor in overall sound environment quality as perceived by adults. In addition, some prior studies have indicated the restorative potential of silence on human health [41]. The duration of each sound was set according to a preliminary test, in which the duration of each sound was set as five levels from 2 min to 6 min with a step of one minute. Ten children were exposed to the soundscapes and were asked their preference of the soundscapes on a five-point scale (from “not at all” to “extremely”). The results showed that a duration of 3 min was most preferred by children. Therefore, 3 min was set as the duration of each sound. Binaural signals of the soundscapes were delivered by computer through headphones (AKG K702) during the experiments.

### 2.3. Measures

In this study, two experiments were designed to explore the restorative benefits of soundscapes on children’s sustained attention and short-term memory, respectively.

In Experiment 1, the sustained attention to response test (SART) was used to test the changes of participants’ attentional capacity, as SART fits the definition of directed attention highlighted in ART. SART is a computer-based go/no-go task which requires participants to withhold behavioral responses to an infrequent and unpredictable target during a period of rapid and rhythmic response to frequent nontargets [34]. The original SART were revised to be more suitable for children to operate, according to preliminary tests. The adapted SART version consisted of 135 digits from one to nine, including 15 targets (i.e., digit 3) and 120 nontargets (except digit 3). The SART was performed using a MATLAB program, and digits were presented to the participant on a computer screen every 1125 ms and remained on the screen for 250 ms. Participants needed to press the space bar every time a nontarget digit was seen and avoid pressing the space bar when viewing the target. Two indices were obtained in this test: (1) Reaction time: The mean response time in milliseconds that participants responded to nontargets; (2) commission errors: The number of times that participants inappropriately pressed the spacebar when the target was presented. The two indices measured response speed and response inhibition, respectively. Notably, participants were asked to give equal weight to responding as quickly as possible and to minimizing commission errors.

In Experiment 2, a digit span test (DST) was used to assess the changes of children’s short-term memory, as there was wide evidence supporting the role of DST as a standardized measure of children’s cognition [42]. DST has a large attentional component, as items must be moved in and out of the focus of attention. During the DST task, a list of random digits (e.g., “8, 3, 6”) was presented on a computer screen at the rate of one every 800 milliseconds. Then, the participant was asked to immediately repeat the digits aloud in the same order they were presented. If they correctly recalled all of the digits, the next list with one digit longer would be presented (e.g., “9, 2, 4, 7”). The length of the list was increased until the participant failed to accurately recall a list of that length on two subsequent occasions. The index obtained in this test was the participant’s digit span, indicating the length of the longest list a participant could remember.

### 2.4. Experimental Design

The experiments were performed in a semi-anechoic chamber in Tianjin University. Participants were accompanied by their parents, who waited in a room outside the chamber during the experiment. Participants took part in the experiment individually, supervised by a researcher.

Before the formal experiment, participants completed the baseline measure of the cognitive test. When the experiment started, they were first given a 5-min oral calculation task, which was assessed to be an effective way to induce attentional fatigue [32]. The difficulty of oral calculation was adapted to children of different age groups to ensure the same difficulty level for all children as much as possible. Specifically, participants aged 7–8 were asked to perform continuous subtraction from a number with three digits with a step of 3 accurately as soon as possible. If he/she did a miscalculation, they would be asked to stop and start their calculation from the beginning. Participants aged 9–10 performed the subtraction from a number with four digits with a step of 7, while participants aged 11–12 performed the subtraction from a number with four digits with a step of 13. Especially, a few children’s oral calculation abilities were obviously better than those of their peers, so in those cases, a more difficult level of calculation was accordingly used. After that, participants were asked to perform the same cognitive test to assess attentional fatigue. The restoration period followed, when participants were exposed to an audiovisual soundscape for 3 minutes without any disturbance. Then, they were asked to perform the cognitive test again to examine whether their cognitive performance returned to normal level. Each experimental unit last around 13 minutes.

In order to control the whole experiment time within one hour, each participant could only experience four of seven soundscapes at most. Therefore, each participant performed the experimental unit four times, as shown in Figure 2. To ensure equal numbers of participants across soundscapes, the four soundscapes for each participant were randomly selected from seven soundscapes by computer. The four soundscapes were also counterbalanced in random order to minimize order effect. In total, each soundscape was experienced by 25 participants at least, which was a moderate sample size for statistical analysis. The sample size was set mostly based on the examples of other studies and our experience. In addition, data saturation was determined when the statistical values were rather steady with the addition of the last few participants. The cognitive baseline levels of participants in different soundscape groups would be examined to ensure no group differences across soundscapes.

Finally, the participants were also asked to provide their demographic information, including age in years and gender.

### 2.5. Data Analysis

The present study used SPSS 25.0 (IBM Corporation, Aemonk, NY, USA) to conduct all the statistical analysis. First of all, the baseline difference across soundscape groups was checked using a nonparametric Kruskal–Willis test, which was the prerequisite before the following analysis. Nonparametric tests were chosen for the following data analysis due to the non-normality of the collected data, which was examined by a Kolmogorov–Smirnov test. In both experiments, Wilcoxon signed-rank tests were first applied to compare children’s cognitive performance before and after each soundscape exposure. Then, the Kruskal–Willis Test was used for the comparison of multiple soundscapes. The change values of variables before and after exposure to soundscapes was calculated as dependent variables, to accommodate individual difference. Additionally, the influence of demographic characteristics on restoration was explored using the Mann–Whitney *U* tests or Kruskal–Wills tests. In all analyses, an alpha value less than 0.05 was used as the criterion to determine significant differences.

## 3. Results

### 3.1. Restoration of Sustained Attention

The baseline levels of sustained attention demonstrated no significant difference among participants across different soundscapes. This was true for both measures: Commission errors and reaction time. The difference in sustained attention before and after the soundscape exposure was tested using a Wilcoxon signed-rank test.

Regarding the reaction time (Table 3), there was a significant improvement after exposure to music, birdsong, fountain sound, and stream sound. Additionally, the bell ring sound also showed a possible trend of reducing reaction time, while no difference in reaction time was shown after exposure to ambient noise and silence. After calculating the change values of sustained attention before and after the soundscape exposure, the Kruskal–Willis test yielded a significant difference between soundscapes in reaction time, 𝜒^2^(6) = 24.647, *p* = 0.000. As shown in Figure 3, boxplots of before–after change values of reaction time were presented, showing mean, median, interquartile range, maximum, and minimum values, with higher change values indicating worse restorative effects. It can be seen that all five soundscapes showed restorative effects on reaction time. Specifically, the fountain sound and stream sound gave rise to the best restorative effects, followed by music and birdsong. Pairwise comparison showed that the change values of reaction time were significant lower for ambient noise than for fountain sound (*p* = 0.004) and stream sound (*p* = 0.042).

Regarding the commission errors (Table 4), there was no significant reduction after each soundscape exposure. However, participants made fewer commission errors after exposure to birdsong and made more commission errors after exposure to music and bell ring, although the difference was not statistically significant. In addition, fountain sound, stream sound, and silence had almost no effects on the commission errors. Notably, a significant increase of commission errors was observed after exposure to ambient noise, that is, ambient noise showed adverse effects on children’s ability of response inhibition, even if the sound pressure level (45 dBA) was lower than other soundscapes. The Kruskal–Willis test also yielded a significant difference between different soundscapes in commission errors, 𝜒^2^(6) = 15.315, *p* = 0.018. As shown in Figure 4, boxplots of before–after change values of commission errors were presented, showing mean, median, interquartile range, maximum, and minimum values, with higher change values indicating worse restorative effects. The results showed that birdsong exhibited the best restorative effects, followed by fountain sound, stream sound, and silence. Pairwise comparison only found a significant difference between birdsong and ambient noise (*p* = 0.013).

In total, fountain sound and water sound demonstrated the best restorative effects on response speed (as measured by reaction time), while birdsong showed the best restorative effects on response inhibition (as measured by commission errors). In addition, bell ring and silence had almost no effects on children’s sustained attention in terms of both response inhibition and response speed. However, ambient noise exhibited significantly adverse effects rather than restorative effects on children’s sustained attention, at least in terms of response inhibition.

As for the influence of personal characteristics on the restoration of children’s sustained attention, Mann–Whitney *U* tests showed that there was no significant difference between boys and girls in change values of both reaction time (*U* = −0.609, *p* = 0.542) and commission errors (*U* = −0.458, *p* = 0.647). Kruskal–Wallis tests also showed no significant influence of age on change values of reaction time (𝜒^2^(2) = 0.909, *p* = 0.635) and commission errors (𝜒^2^(2) = 0.882, *p* = 0.643). However, results indicated that participants who conducted more commission errors at baseline period performed better after the soundscape exposure, relative to participants who conducted fewer commission errors at baseline period (*U* = −1.991, *p* = 0.047). However, this difference between baseline levels was not achieved in the restoration of reaction time (𝜒^2^(2) = 2.835, *p* = 0.242).

### 3.2. Restoration of Short-Term Memory

The baseline scores of DST demonstrated no significant difference between the soundscapes. The Wilcoxon signed-rank test was used to assess the change values of digit span before and after exposure to the soundscape stimuli, as shown in Table 5. It can be seen that after exposure to music, birdsong, fountain sound, and steam sound, children’s digit span was substantially improved, while bell ring, ambient noise, and silence showed no actual restorative effects.

The change values of digit span before and after exposure to different soundscapes were compared using Kruskal–Wallis tests. As shown in Figure 5, boxplots of before–after change values of digit span were presented, showing mean, median, interquartile range, maximum, and minimum values, with higher change values indicating better restorative effects. The results showed a main effect of soundscapes on the restoration of children’s short-term memory (𝜒^2^(6) = 19.876, *p* = 0.003). Among the seven soundscapes, children’s digit span improved to a significantly greater extent after exposure to fountain sound, relative to bell ring (*p* = 0.029) and ambient noise (*p* = 0.018), as indicated by a pairwise comparison. In addition, stream sound also showed a relatively good restorative effect on short-term memory, followed by music and birdsong, as shown in Figure 5.

Regarding the influence of children’s demographic characteristics, the nonparametric analysis showed that the change values of children’s digit span did not differ between genders (*p* = 0.962), ages (*p* = 0.128), and baseline levels (*p* = 0.149).

## 4. Discussion

In this study, two experiments were conducted to examine the effects of restorative soundscapes on children’s cognitive performance. In Experiment 1, children’s sustained attention was measured by SART. Although the bell ring did not show any restorative effects on children’s sustained attention, we did find evidence for the restorative effects of music, birdsong, fountain sound, and stream sound on children’s response speed, as measured by reaction time. These findings not only provide empirical support for our previous study that explored restorative soundscapes based on children’s perceptions [22], but also extend understanding of other works with adults, which only indicated the cognitive benefits of natural sounds in comparison with artificial sounds [26]. However, those soundscapes did not show significant restorative effects on response inhibition, as measured by commission errors. A possible reason was that response speed was reported to be negatively related to the performance of response inhibition [43], which was also confirmed in this study at the baseline levels (*p* = 0.000). Therefore, the results indicated that acute exposure to restorative soundscapes is not sufficient to cause positive effects on those two separate cognitive performances simultaneously. In addition, change values in reaction time did not correlate with changes in commission errors after soundscape exposure (*p* = 0.918), suggesting that the observed improvements in reaction time were not driven by the decrease in commission errors. In Experiment 2, short-term memory was measured by DST, in which digit span was used as the index of short-term memory. The results demonstrated that a brief exposure to fountain sound and stream sound could indeed help children to recover from a state of induced cognitive fatigue. As perceived restorative soundscapes, music and birdsong also have positive effects on children’s short-term memory, although the effects were not significant. These results suggested that water soundscapes, such as fountain sound and stream sound, could be practically designed and played back during the intervals of classes for children who need various cognitive resources, including both sustained attention and short-term memory. Especially, music and birdsong could be mainly used for children who need restoration and improvement of response speed.

Comparing the change values of those three cognitive indices, it can be seen that the restorative effects of some soundscapes (such as fountain sound, stream sound, music and birdsong) were actually achieved in both children’s response speed and digit span, rather than response inhibition. This might be explained according to previous studies on child psychology development, which suggested that response speed and memory span are usually viewed as a simple cognitive progress and could be improved easily in a short period. Response inhibition, however, is one of the integrative executive functions and a comparative complex progress for goal-directed behavior [44]. Therefore, the results in this study indicated that restorative soundscapes may affect different cognitive processes to a various extent. Although the restorative effects of soundscapes on children’s basic cognitive functioning need further study with a larger sample size, this line of reasoning might lead to a number of intriguing research possibilities on various cognitive capabilities.

In addition, consistent with our hypothesis, the five potential restorative soundscapes as perceived by children in a previous study (i.e., music, birdsong, fountain sound, bell ring, and stream sound) showed significantly different effects on children’s cognitive performance [22]. Generally, fountain sound and stream sound yielded the best restorative effects, followed by music and birdsong, while bell ring showed possible adverse effects instead. The results could be partly explained by the psychoacoustic characteristics of soundscapes. On one hand, as shown in Table 2, the psychoacoustic parameters (fluctuation strength, sharpness, and roughness) of music sound, birdsong, and bell ring changed to a greater extent compared with the fountain sound and stream sound after combination with the ambient noise. This indicated that children’s perceptions of fountain sound and stream sound were less influenced by the noise relative to other sounds. This was also approved by previous studies, which suggested water sounds were more appropriate than other sounds for masking noise [45]. On the other hand, it is interesting to note that the soundscapes which were perceived as more restorative, such as music and birdsong, did not necessarily show better actual restorative effects compared to fountain sound and stream sound, which were perceived as less restorative by children, as shown by the PRSS-C evaluation in Table 2. One possible reason is that music sound and birdsong have much greater fluctuation strength than fountain sound and stream sound, and may thus lead to higher mood vibrancy [46]. By contrast, water sounds with smaller fluctuation strength might make it much easier for children to restore calm and stability during the cognitive task after the soundscape exposure. Another alternative explanation could be that children were more relaxed after exposure to music, which substantially lowered children’s arousal levels, thereby reducing their cognitive performance instead. This interpretation is consistent with the Yerkes–Dodson Law [47]. In addition, the bell ring showed the least restorative effects on children’s cognitive performance among the five potential restorative soundscapes. This might be because of its extremely high sharpness, which is caused by high-frequency components in the sound. Although sharpness was found to be essential for preference, it was not helpful in enhancing the calmness [45]. Moreover, it is important to note that there were only six soundscapes used in this study, and the psychoacoustic characteristics of those soundscapes were only used to partly explain the possible reason for the results. Future studies are still needed to confirm the correlation between the psychoacoustic parameters and soundscapes’ cognitive effects on children.

Furthermore, this study provides more empirical support for previous research, which demonstrated that ambient noise in classrooms had significant adverse effects on children’s cognitive performance [37,48,49]. It was apparent that the current noise level (45 dBA) is not sufficient to achieve the desired effects on children’s cognition, thus raising the need for more effective noise reduction measures. However, there was only one fixed background noise level to simulate the real-life sound environment in school classrooms in this study. Therefore, more future studies should be performed to investigate the influence of different background noise levels on the restorativeness of soundscapes. It is also interesting to note that exposure to silence did not facilitate recovery on children’s cognitive performance, which was quite different from previous studies with adults that suggested quietness has positive effects on adults’ health and well-being [41,50]. Therefore, noise reduction is not necessarily sufficient to define a restorative acoustic environment in school classrooms. The environmental design of restorative soundscapes, such as fountain sound and stream sound, is essential for children’s cognitive recovery.

Finally, although we found no significant effects of gender and age on children’s cognitive restoration, future studies might be required to investigate whether children’s personal characteristics may alter the restorative effects of soundscapes. However, it is worthy to note that children’s self-reported normal hearing and vision were used as the measures of their perceptual abilities in this study. In order to exclude the possible influence of the individual difference in perceptual accuracy, a more accurate hearing and vision test should be performed in future studies. Moreover, we found that children with a lower baseline level of response inhibition were more restored after exposure restorative soundscapes than those of higher baseline level. This result is in line with previous studies, which suggested that acoustic environments affect cognitive performance differently depending on individual differences in cognition baseline [51]. Although the results are less conclusive and need examination through more research, they may imply that more attention should be given to the design of restorative soundscapes for children with poorer cognitive abilities.

## 5. Conclusions

In a simulated situation of a school classroom, two experiments were carried out to examine the restorative effects of soundscapes on children’s cognitive performance. Based on children’s performance of sustained attention and short-term memory, the following conclusions could be drawn.

1. Among the seven soundscapes, water sound and fountain sound showed the best restorative effects on children’s cognitive performance, followed by music and birdsong. Bell ring and silence showed no significant restorative effects, while ambient noise showed adverse effects.

2. Regarding children’s sustained attention, restorative soundscapes showed benefits in terms of response speed, except for bell ring. However, children’s capability of response inhibition did not differ significantly before and after exposure to those soundscapes.

3. Regarding children’s short-term memory, only fountain sound and stream sound showed significant restorative effects. Music and birdsong also had positive effects on short-term memory; however, the effects were not significant.

4. Children’s gender and age had no influence on their restoration of cognitive performance. In addition, restorative soundscapes might have more restorative function on the response inhibition of children with lower attentional baseline level than those with higher attentional baseline level.

## Figures and Tables

**Figure 1 ijerph-16-00293-f001:**
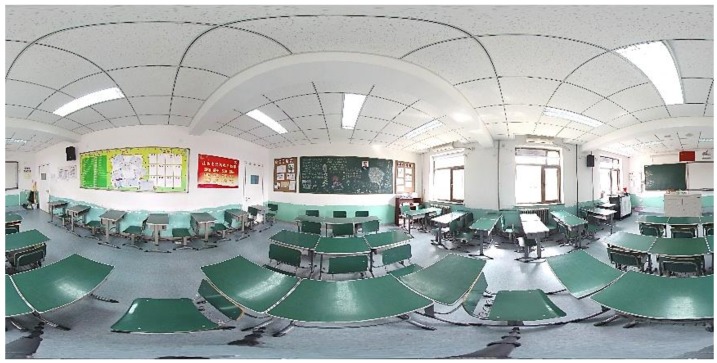
The panoramic view of a school classroom.

**Figure 2 ijerph-16-00293-f002:**
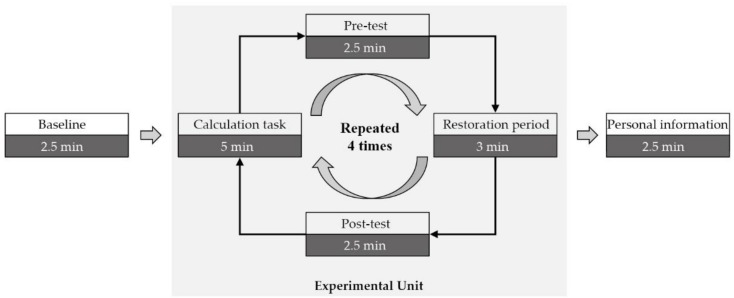
Diagram of experimental procedure.

**Figure 3 ijerph-16-00293-f003:**
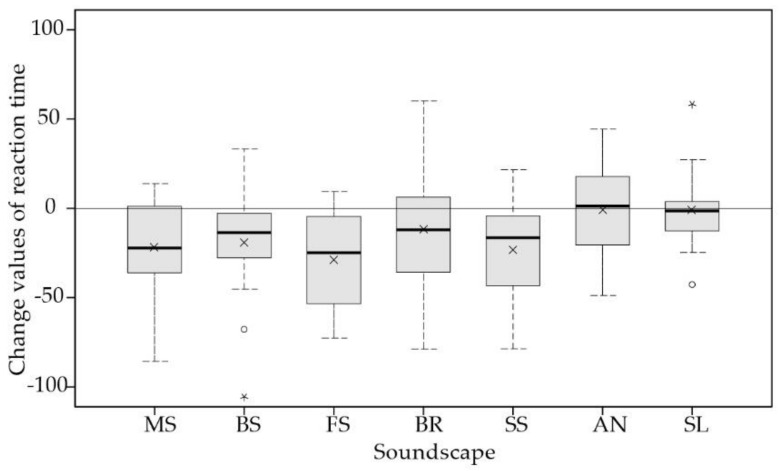
Boxplots of change values of reaction time between soundscapes, showing median, interquartile range, maximum, and minimum values. The crosses represent mean values. MS, BS, FS, BR, SS, AN, and SL represent music, birdsong, fountain sound, bell ring, stream sound, ambient noise, and silence, respectively.

**Figure 4 ijerph-16-00293-f004:**
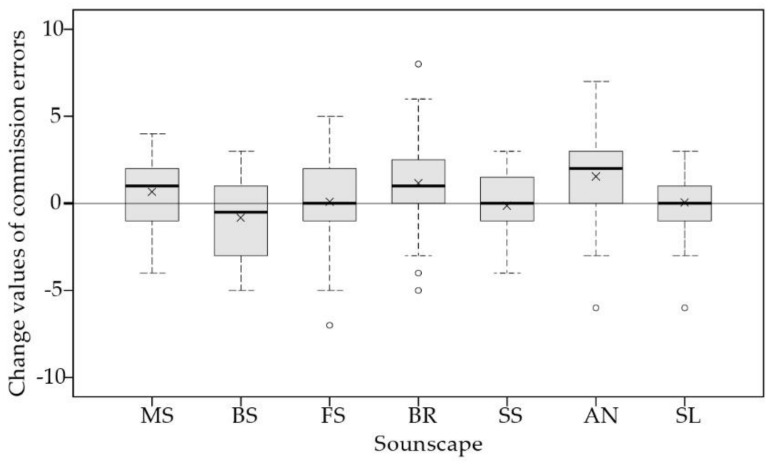
Boxplots of change values of commission errors between soundscapes, showing median, interquartile range, maximum, and minimum values. The crosses represent mean values. MS, BS, FS, BR, SS, AN, and SL represent music, birdsong, fountain sound, bell ring, stream sound, ambient noise, and silence, respectively.

**Figure 5 ijerph-16-00293-f005:**
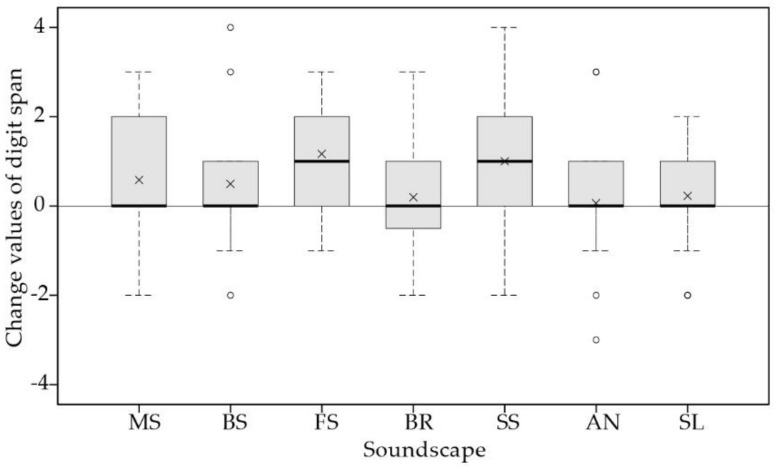
Boxplots of change values of digit span between soundscapes, showing median, interquartile range, maximum, and minimum values. The crosses represent mean values. MS, BS, FS, BR, SS, AN, and SL represent music, birdsong, fountain sound, bell ring, stream sound, ambient noise, and silence, respectively.

**Table 1 ijerph-16-00293-t001:** Demographics characteristic of the participants in this study.

Variable	Characteristic	Experiment 1 (*n*)	Experiment 2 (*n*)
Gender	Boy Girl	23 23	22 23
Age (years)	7–8 9–10 11–12	14 16 16	14 15 16
Baseline (CE)	≤9 ≥10	22 24	
Baseline (RT)	<450 450–500 >500	18 13 15	
Baseline (DS)	≤7 ≥8		23 22
Total		46	45

CE = Commission errors (number); RT = Reaction time (millisecond); DS = Digit span (length).

**Table 2 ijerph-16-00293-t002:** Psychoacoustic characteristics and restorative evaluation of sound signals and combined soundscapes.

	Loudness (sone GF)		Fluctuation Strength (vacil)		Sharpness (acum)		Roughness (asper)		PRSS-C
	Sound signal	Soundscape	Sound signal	Soundscape	Sound signal	Soundscape	Sound signal	Soundscape	Soundscape
MS	8.54 (1.65)	14.80 (2.27)	0.058 (0.043)	0.046 (0.024)	1.31 (0.25)	2.05 (0.30)	0.74 (0.24)	1.53 (0.36)	3.71 (0.72)
BS	4.30 (1.23)	11.50 (2.11)	0.058 (0.044)	0.040 (0.022)	2.66 (0.60)	2.41 (0.33)	0.17 (0.22)	1.60 (0.43)	3.43 (0.78)
FS	7.92 (0.29)	13.00 (1.30)	0.005 (0.006)	0.012 (0.011)	2.76 (0.06)	2.79 (0.15)	1.61 (0.10)	1.92 (0.25)	3.29 (0.80)
BR	5.10 (1.90)	12.00 (2.10)	0.013 (0.008)	0.032 (0.020)	4.43 (1.02)	2.89 (0.58)	0.24 (0.20)	1.63 (0.43)	3.17 (0.82)
SS	7.14 (0.71)	13.40 (1.59)	0.031 (0.009)	0.033 (0.012)	2.39 (0.14)	2.54 (0.21)	2.34 (0.25)	2.76 (0.30)	3.12 (0.82)
AN	6.28 (1.28)		0.028 (0.018)		2.20 (0.31)		0.95 (0.42)		2.55 (0.63)

Mean values and standard deviation (SD) of psychoacoustics parameters for each sound signal and soundscape were presented in the table. MS = Music; BS = Birdsong; FS = Fountain sound; BR = Bell ring; SS = Stream sound; AN = Ambient noise. PRSS-C = Perceived restorative soundscape scale for children.

**Table 3 ijerph-16-00293-t003:** The statistical scores of reaction time before and after exposure to each soundscape.

Soundscape	Before		After		*Z* (w)	*p*
	M (SD)	Median (IQR)	M (SD)	Median (IQR)		
MS	480.29 (83.21)	481.65 (105.4)	458.34 (77.41)	448.10 (118.4)	−3.264	0.001
BS	481.97 (93.35)	449.80 (110.6)	462.87 (80.72)	453.15 (98.0)	−3.238	0.001
FS	497.64 (86.14)	489.75 (161.5)	469.11 (79.60)	454.90 (127.8)	−4.203	0.000
BR	478.54 (88.35)	480.60 (117.0)	466.95 (81.25)	465.00 (129.4)	−1.898	0.058
SS	476.57 (88.37)	460.90 (138.3)	453.20 (78.51)	444.40 (96.2)	−3.628	0.000
AN	480.98 (77.34)	476.30 (114.4)	479.79 (86.04)	476.05 (124.2)	−0.063	0.949
SL	459.53 (86.22)	450.90 (113.3)	458.46 (90.02)	454.50 (101.3)	−0.902	0.367

MS = Music; BS = Birdsong; FS = Fountain sound; BR = Bell ring; SS = Stream sound; AN = Ambient noise; SL = Silence; IQR = Interquartile range.

**Table 4 ijerph-16-00293-t004:** The statistical scores of commission errors before and after exposure to each soundscape.

Soundscape	Before		After		*Z* (w)	*p*
	M (SD)	Median (IQR)	M (SD)	Median (IQR)		
MS	8.85 (3.52)	8.50 (5.3)	9.50 (3.81)	10.00 (5.8)	−1.488	0.137
BS	9.69 (3.88)	10.50 (4.8)	8.85 (3.96)	8.50 (6.5)	−1.610	0.107
FS	9.73 (3.60)	10.00 (6.0)	9.72 (3.59)	10.00 (6.0)	−0.062	0.758
BR	8.52 (4.47)	9.00 (8.0)	9.67 (4.02)	10.00 (7.0)	−1.961	0.050
SS	9.37 (4.19)	10.00 (7.0)	9.22 (4.07)	10.00 (7.0)	−0.308	0.951
AN	8.42 (4.26)	8.00 (8.3)	10.12 (3.55)	10.00 (5.3)	−2.687	0.007
SL	9.81 (3.49)	10.00 (4.3)	10.00 (3.40)	10.00 (5.0)	−0.285	0.775

MS = Music; BS = Birdsong; FS = Fountain sound; BR = Bell ring; SS = Stream sound; AN = Ambient noise; SL = Silence; IQR = Interquartile range.

**Table 5 ijerph-16-00293-t005:** The statistical scores of digit span before and after exposure to each soundscape.

Soundscape	Before		After		*Z* (w)	*p*
	M (SD)	Median (IQR)	M (SD)	Median (IQR)		
MS	7.65 (1.41)	7.00 (2.0)	8.23 (1.38)	8.50 (2.0)	−2.251	0.024
BS	7.77 (1.34)	8.00 (2.0)	8.27 (1.34)	8.00 (2.0)	−2.053	0.040
FS	7.20 (1.53)	7.00 (5.0)	8.36 (1.80)	8.00 (1.0)	−3.699	0.000
BR	7.67 (1.47)	8.00 (2.0)	7.85 (1.70)	7.00 (2.0)	−0.700	0.484
SS	7.46 (0.99)	7.00 (1.0)	8.46 (1.50)	8.00 (3.0)	−3.214	0.001
AN	7.88 (1.48)	8.00 (2.0)	7.96 (1.84)	8.00 (4.0)	−0.254	0.799
SL	7.68 (1.44)	8.00 (2.0)	7.92 (1.44)	8.00 (2.0)	−1.039	0.299

MS = Music; BS = Birdsong; FS = Fountain sound; BR = Bell ring; SS = Stream sound; AN = Ambient noise; SL = Silence; IQR = Interquartile range.

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
