# Peer review of "Restorative Effects of Classroom Soundscapes on Children’s Cognitive Performance"

_ijerph, 2019, doi:10.3390/ijerph16020293_

Round 1

Reviewer 1 Report

 The paper “Restorative Effects of Classroom Soundscapes on Children’s Cognitive Performance” presents an interesting experiment and sound results, which might be very helpful for future applications. In order to improvve the clarity of the paper, this reviewer suggest to better address the following points:

-line 44: please clarify specifically the references 15, 16, 17, 18, 19 at this point. Do they report the same results? Based on the same experimental conditions?

- a critical point of view is missing in presenting the Perceived Restorativeness Soundscape Scale (PRSS).

-lines 45-69: how does the presentation methodology of the soundscape and the cognitive task typology influence the results? How does the author’s study relate/differ to the previous methodologies?

-self-reported normal hearing and vison have great uncertainty thus a more accurate test should be performed. A discussion on this limitation of the study should be presented

-table 2: what do the numbers in brackets represent? Please add the JND values of the psychoacoustic parameters.

-please add more information on why silence has been chosen as control stimulus.

-line 159: how was performed the preliminary test? Was it the same as S/N test evaluation?

-lines 161- was there any headphones equalization applied?

-line 172: how was the timing decided?

- line 180- how were the answers gathered?

Line 198: how?

-line 209: “adequate sample size for statistical analysis.”…How was this assessed?

Figure 2. was the calculation task always the same? How does the memorization of the results influence the outcome?

Table 3, 4, 5: z(w)? P should be lower case letter.

Figure 3, 4, 5 should be better explained in the text and not only in the figure caption

Line 250-participants -p lower case

Lines 312-313 and 341-358 It results necessary to clarify in a more systematic way the correlation with the psychoacoustic characteristics of the soundscape sounds?

Line 355: “It might because…” seems like a verb is missing.

The discussion and conclusions should include also possible practical solutions of the presentation of these restorative sounds. How is expected to change the effect if there are also other students in the classroom. Does the background noise affect substantially the results?

Author Response

Response to Reviewer 1 Comments

Point 1: line 44: please clarify specifically the references 15, 16, 17, 18, 19 at this point. Do they report the same results? Based on the same experimental conditions?

Response 1: Reviewer’s comment is very reasonable. Those references reported different results based on different experimental conditions. According to the comment, more detailed clarification was added and marked in red colour in Page 2 Line 44-47.

Point 2: a critical point of view is missing in presenting the Perceived Restorativeness Soundscape Scale (PRSS).

Response 2: According to the comment, more detailed information about the construct and result of PRSS was added and marked in red colour in Page 2 Line 50-55.

Point 3: lines 45-69: how does the presentation methodology of the soundscape and the cognitive task typology influence the results? How does the author’s study relate/differ to the previous methodologies?

Response 3: Up to now, although emerging research has been carried out on restorative soundscape, a systematic methodology of this research area is still lacking. Those previous empirical studies applied quite different experiment stimuli, presentation methods and measurements, thereby resulting in inconsistent outcomes. However, it is noteworthy that the research using audio-visual stimuli usually resulted in more restorative effects than those using sound stimuli only. In addition, various cognitive tests regarding attention and memory were mainly used as sensitive measures of cognitive restoration. Detailed information was added and marked in red colour in Page 2 Line 74-79.

Therefore, we selected the soundscape presentation method and cognitive tasks according to following criteria: 1) soundscape should be presented to simulate the real classroom environment as much as possible, a VR technology was thereby applied to provide audio-visual stimuli; and 2) attentional and memory tests that were prone to be affected by soundscapes in previous studies was selected, but they were further revised to adapt to children group.

Point 4: self-reported normal hearing and vision have great uncertainty thus a more accurate test should be performed. A discussion on this limitation of the study should be presented

Response 4: Reviewers’ comment is totally right. This is indeed a limitation of our study. A more accurate hearing and vision test should be performed to exclude the possible influence of the individual difference in perceptual accuracy. A discussion on this has been presented in the revised manuscript and marked in red colour in Page 12 Line 439-442.

Point 5: table 2: what do the numbers in brackets represent? Please add the JND values of the psychoacoustic parameters.

Response 5: The numbers in brackets represent the standard deviation (SD). Detailed information was added in the table caption and marked in red colour in Page 5 Line 184-185.

As reported in a previous study, the JNDs of loudness, fluctuation strength, sharpness, and roughness were 0.5 sone, 0.012 vacil, 0.08 acum, and 0.04 asper, respectively. Detailed information was added and marked in red colour in Page 5 Line 176-181.

Reference:

You J, Jeon J Y. Just noticeable differences in sound quality metrics for refrigerator noise. Noise Control Engineering Journal. 2008, 56(6), 414-424.

Point 6: please add more information on why silence has been chosen as control stimulus.

Response 6: Silence was widely reported to be an important factor in overall sound environment quality as perceived by adults. In addition, some prior studies have indicated the restorative potential of silence on human health. Detailed information was added and marked in red colour in Page 5 Line 190-192.

Point 7: line 159: how was performed the preliminary test? Was it the same as S/N test evaluation?

Response 7: Yes, it was almost the same as S/N test. Detailed information about the test was added and marked in red colour in Page 5 Line 192-197.

Point 8: lines 161- was there any headphones equalization applied?

Response 8: Before the combination, the sound signals were normalized to the average sound level of 50dBA (45dBA for ambient noise) using a Norsonic Nor140 Class 1 sound level meter coupled to a reference class headphone AKG K702, whose highly sensitivity could ensure the accuracy of the emitted signal with respect to the real sound. Both the adjustment of sound pressure level and the combination of soundscapes were processed by Adobe Audition software. Detailed information was added and marked in red colour in Page 5 Line 170-174.

Point 9: line 172: how was the timing decided?

Response 9: The timing was decided according to the standard SART program which was used in many previous studies, which suggested 1125ms was enough time for people to respond, and 250ms was an appropriate time to show the digit clearly to participants.

Point 10: line 180- how were the answers gathered?

Response 10: After the presentation of digits, the participant was asked to immediately repeat the digits aloud in the same order they were presented. Detailed information was added and marked in red colour in Page 6 Line 220-222.

Point 11: Line 198: how?

Response 11: The difficulty of oral calculation was adapted to children of different age groups to ensure the same difficulty level for all children as much as possible. Specifically, participants aged 7-8 were asked to perform continuous subtraction from a number with three digits with a step of 3 accurately as soon as possible. If he/she did a miscalculation, they would be asked to stop and start their calculation from the beginning. Participants aged 9-10 performed the subtraction from a number with four digits with a step of 7, while participants aged 11-12 performed the subtraction from a number with four digits with a step of 13. Especially, a few children’s oral calculation abilities were obviously better than their peers, then a more difficult level of calculation was accordingly used. Detailed information was added and marked in red colour in Page 6 Line 233-241.

Point 12: line 209: “adequate sample size for statistical analysis.”…How was this assessed?

Response 12: The sample size was set mostly based on the examples of other studies and our experience. In addition, the data saturation was determined when the statistical values were rather steady with the addition of the last few participants. Detailed information was added and marked in red colour in Page 7 Line 252-254.

Point 13: Figure 2. was the calculation task always the same? How does the memorization of the results influence the outcome?

Response 13: Each calculation task was started from different numbers to avoid the memorization of the results. In addition, the calculation task was used to induce cognitive fatigue, and the calculation outcome was not what we focused on.

Point 14: Table 3, 4, 5: z(w)? P should be lower case letter.

Response 14: z(w) is the same as t values in the parametric analysis. In addition, according to the comment, P was changed to lower case letter p in three tables.

Point 15: Figure 3, 4, 5 should be better explained in the text and not only in the figure caption

Response 15: Reviewer’s comment is completely right. More detailed information about Figure 3, 4, 5 was added in the text and marked in red colour in Page 7-8 Line 283-285, Page 8 Line 305-307 and Page 10 Line 343-345, respectively.

Point 16: Line 250-participants -p lower case

Response 16: According to the comment, P was changed to lower case letter p in Page 8 Line 298.

Point 17: Lines 312-313 and 341-358 It results necessary to clarify in a more systematic way the correlation with the psychoacoustic characteristics of the soundscape sounds?

Response 17:  Reviewer’s comment is very valuable, and the result was explained and clarified in more systematic way in Page 11-12 Line 399-423 as marked in red colour. It is important to note that, there were only six soundscapes used in this study, and the psychoacoustic characteristics of those soundscapes were only used to partly explain the possible reason for the results. Future studies are still needed to confirm the correlation between the psychoacoustic parameters and soundscapes’ cognitive effects on children.

Point 18: Line 355: “It might because…” seems like a verb is missing.

Response 18: According to the comment, the sentence was revised as “It might be because of its extremely high sharpness” in Page 12 Line 417.

Point 19: The discussion and conclusions should include also possible practical solutions of the presentation of these restorative sounds. How is expected to change the effect if there are also other students in the classroom. Does the background noise affect substantially the results?

Response 19: Reviewer’s comment is completely right. The Results in this study suggested that water soundscapes such as fountain sound and stream sound could be practically designed and played back during the intervals of classes for children who need various cognitive resources including both sustained attention and short-term memory. Especially, music and birdsong could be mainly used for children who need restoration and improvement of response speed. In addition, more attention should be paid on the design of restorative soundscapes for children with poorer cognitive abilities. According to the comment, the detailed discussion above was added and marked in red colour in Page 11 Line 378-383 and Line 446-448.

Regarding the influence of background noise on the results, this study showed that the current noise level (45dBA) is not sufficient to achieve the desired effects on children’s cognition, thus raising the need for more effective noise reduction measures. Detailed information was added and marked in red colour in Page 12 Line 426-430.

Reviewer 2 Report

This study investigated the restorative effect of classroom soundscape changes on cognitive performance of children. The authors set the various soundscape sources to background noise and measured attention and short-term memory using the sustained attention to response test (SART) and the digit span test (DST) method. This study is considered to have originality in that it uses methodology to evaluate cognitive performance according to various soundscape sources in VR environment. However, the experimental method and the statistical analysis seem to be needed further supplementation.

First, it is necessary to secure the normality of the subjects. The results presented in this paper compare the effects of soundscape sources with their relative magnitudes, but it is necessary to secure the normality of the subjects and present the magnitude of the actual effect through parametric analysis. It is also necessary to verify the age and gender once the normality has been secured.

The following are minor comments on the manuscript.

Pate 3, Table 1: It is necessary to add the units for Baseline (RT)

Line 141: It is necessary to add more explanation about each sound source. For example, in the case of Fountain sound, there are previous studies that have different soundscapes depending on the kind of fountain. Therefore, a detailed explanation is needed in addition to frequency analysis for each sound source.

Author Response

Response to Reviewer 2 Comments

Point 1: It is necessary to secure the normality of the subjects. The results presented in this paper compare the effects of soundscape sources with their relative magnitudes, but it is necessary to secure the normality of the subjects and present the magnitude of the actual effect through parametric analysis. It is also necessary to verify the age and gender once the normality has been secured.

Response 1: Reviewer’s comment is completely right. The relative magnitudes of actual effects were used in order to accommodate individual difference, and we have checked the normality of the magnitudes using Kolmogorov-Smirnov test. However, the results showed the data was not normally distributed. Therefore, a non-parametric Kruskal-Willis test was used for the comparison of multiple soundscapes. In addition, the sample size of each age and gender was paralleled as shown in Table 1.

Point 2: Page 3, Table 1: It is necessary to add the units for Baseline (RT)

Response 2: Reviewers’ comment is totally right. The units for Baseline (RT) is a millisecond, and it was added and marked in yellow colour in Page 4 Line 130.

Point 3: Line 141: It is necessary to add more explanation about each sound source. For example, in the case of Fountain sound, there are previous studies that have different soundscapes depending on the kind of fountain. Therefore, a detailed explanation is needed in addition to frequency analysis for each sound source.

Response 3: Reviewer’s comment is completely right. Specifically, a typical classical piano music, named “Souvenirs d'enfance”, was selected as the music stimulus, which was a piece of soft and relaxing music without lyrics. In addition, a series of chirps of sparrows and other few bird species were recorded in a park in Tianjin and used as the birdsong stimulus. The fountain sound was produced by an upward shallow jet and water falling along all sides of a square stone column, while the stream sound was recorded nearby a downwards stream consisted with slightly tilted stone steps and slowly flowing water. The sound of bell ring was produced by irregularly striking a wind chime which was constructed from a series of suspended mental bells. The detailed explanation was added and marked in yellow colour in Page 4 Line 146-153.

Reviewer 3 Report

Review of manuscript: Restorative Effects of classroom soundscapes on children’s cognitive performance

General comments

The manuscript reports on the effect of seven soundscapes on the cognitive performance of children aged 8 to 12 years. Two basic cognitive abilities were considered (sustained attention and short-term memory) which were deemed essential for performing academic activities. Results indicated restorative effects of natural soundscapes and an adverse effect of classroom ambient noise.

It was a pleasure to review this manuscript. It is well structured, and interesting to read. The rationale is well motivated and the introduction gives as essential, yet coherent, background. I believed the manuscript represents a timely contribution, which deepens the current knowledge on the effects of the classroom acoustic environment on the pupils.

I have only some minor remarks, which are detailed in the following list:

1\ page 3, line 103: Did you also control that the participants had a typical development?

2\ page 4, line 131: The reason #2 is not clear for me: none of the environmental sounds is typical of a school and potentially any soundscape could be play back in a classroom. Could you please clarify the sentence?

3\ pages 4-5, lines 144-155: It would be of interest to add the psychoacoustic characteristics of the actual soundscapes played back during the experiment, including both ambient noise and the sound signal. Given the selected S/N (+5), the characteristics of the final soundscapes will probably differ from those reported in Table 2.

4\ page 5, line 153 (and subsequent occurrences): It is not common to give the S/N in dBA. Signal and noise levels are measured in dB(A), and their difference is usually reported in dB.

5\ page 5, line 183: Replace “DSF” with “DST”.

6\ page 5, lines 180-188: Please specify how the children performed the DST: how were the digits presented? how the participants answered (orally, using a keyboard…)?

7\ page 5, line 198: Please specify how the oral calculation task was adapted to the different groups.

8\ page 8, line 286: Replace “DSF” with “DST”.

Author Response

Response to Reviewer 3 Comments

Point 1: page 3, line 103: Did you also control that the participants had a typical development?

Response 1: Yes. We excluded a few participants according to their basic cognitive performance. Actually, there were 95 participants took part in the experiments totally, but four of them were excluded during the data analysis, because their baseline cognitive scores were outliers. Detailed information was added and underlined in Page 3 Line 115-116.

Point 2: page 4, line 131: The reason #2 is not clear for me: none of the environmental sounds is typical of a school and potentially any soundscape could be play back in a classroom. Could you please clarify the sentence?

Response 2: Reviewers’ confuse is totally understandable. Among the five potential restorative soundscapes, music could be played back through school broadcasting system during recess time. Birdsong, fountain sound and stream sound could be added through schoolyard landscape design, because the sounds in the schoolyard are also an important background sound resources that could be heard from classroom, especially when the window are open after classes. Bell ring could be used as a school bell sound.

Reviewer is totally right about that potentially any soundscape could be played back in a classroom through loudspeakers, but we want to add some soundscapes which are coherent with the classroom environments and could be easily added through corresponding soundscape design. It might be quite weird to add some sounds such as ocean wave sound, since it is totally incoherent with the classroom settings. A more clear explanation was underlined in Page 4 Line 144-146.

Point 3: pages 4-5, lines 144-155: It would be of interest to add the psychoacoustic characteristics of the actual soundscapes played back during the experiment, including both ambient noise and the sound signal. Given the selected S/N (+5), the characteristics of the final soundscapes will probably differ from those reported in Table 2.

Response 3: Reviews’ suggestion is very interesting and valuable. Psychoacoustic characteristics of actual soundscapes was added in Table 2. It is interesting to find that the fluctuation strength, sharpness and roughness of music sound, birdsong and bell ring changed to a greater extent relatively to the fountain sound and stream sound after combination with the ambient noise. More detailed information and corresponding discussion was added and underlined in Page 11 Line 400-406.

Point 4: page 5, line 153 (and subsequent occurrences): It is not common to give the S/N in dBA. Signal and noise levels are measured in dB(A), and their difference is usually reported in dB.

Response 4: Reviewer is totally right. All S/N in dBA have been changed to dB in the revised manuscript and underlined in Page 5 Line 168 and Line 170.

Point 5: page 5, line 183: Replace “DSF” with “DST”.

Response 5: According to the comment, all DSF” in the manuscript has been replaced with “DST”.

Point 6: page 5, lines 180-188: Please specify how the children performed the DST: how were the digits presented? how the participants answered (orally, using a keyboard…)?

Response 6: During the DST task, a list of random digits (e.g., “8, 3, 6”) was presented on a computer screen at the rate of one every 800 milliseconds. Then the participant was asked to immediately repeat the digits aloud in the same order they were presented. Detailed information was added and underlined in Page 6 Line 220-222.

Point 7: page 5, line 198: Please specify how the oral calculation task was adapted to the different groups.

Response 7: Specifically, participants aged 7-8 were asked to perform continuous subtraction from a number with three digits with a step of 3 accurately as soon as possible. If he/she did a miscalculation, they would be asked to stop and start their calculation from the beginning. Participants aged 9-10 performed the subtraction from a number with four digits with a step of 7, while participants aged 11-12 performed the subtraction from a number with four digits with a step of 13. Especially, a few children’s oral calculation abilities were obviously better than their peers, then a more difficult level of calculation was accordingly used. Detailed information was added and underlined in Page 6 Line 233-241.

Point 8: page 8, line 286: Replace “DSF” with “DST”.

Response 8: According to the comment, all DSF” in the manuscript has been replaced with “DST”.

Round 2

Reviewer 1 Report

This reviewer has carefully revised the second version of the article, which has been substantially improved. The authors have answered to all the issues raised by the reviewer.

Reviewer 2 Report

I believe the author has adequately supplemented the manuscript of the reviewer's comments. The authors made clear the limitations of this study and provided a good direction for future researches. Therefore, this study is appropriate for publication in the IJERPH Journal.